# Leveraging Mutual Information for Asymmetric Learning under Partial Observability

**Hai Nguyen[1], Long Dinh[2], Robert Platt[1], Christopher Amato[1]**
[1] Khoury College of Computer Sciences, Northeastern University, Boston, MA, United States
[2] Horus AI, Hanoi, Vietnam
https://sites.google.com/view/mi-asym-pomdp

**Abstract:** Even though partial observability is prevalent in robotics, most reinforcement learning studies avoid it due to the difficulty of learning a policy that can efficiently memorize past events and seek information. Fortunately, in many cases, learning can be done in an asymmetric setting where states are available during training but not during execution. Prior studies often leverage the state to indirectly influence the training of a history-based actor (actor-critic methods) or a history-based critic (value-based methods). Instead, we propose using state-observation and state-history mutual information to improve the agent's architecture and ability to seek information and memorize efficiently through intrinsic rewards and an auxiliary task. Our method outperforms strong baselines through extensive experiments and achieves successful sim-to-real transfers to a real robot.

**Keywords:** Partial Observability, Mutual Information, Asymmetric Learning

## 1 Introduction

While partial observability is ubiquitous in robotics [1, 2], most reinforcement learning (RL) robotics research avoids it due to the difficulty of training a policy that can effectively memorize past information and perform information-gathering actions. Fortunately, full or near-full observability during training is possible in many real-world domains using a simulator or a generative environment model. In the literature, this is the asymmetric learning setting [3, 4, 5, 6, 7, 8], in which privileged information such as states can be available *during training*. Previous work in this setting often learns a state-based critic to *indirectly* influence the learning process of a) a history-based actor [3, 9] using actor-critic RL to output actions directly, or b) a history-based critic [6] in value-based RL to extract actions from. In contrast, using an information-theoric approach, we instead use the state to generate intrinsic rewards and optimize an auxiliary task, shaping the agent's behavior *directly*.

Specifically, this study proposes to use the mutual information between the state and the history during training to motivate active information gathering and to improve task-relevant memorization. Our intuition is that active information gathering mainly involves discovering actions that help the agent reveal more about the environment's state, gradually moving from partial to full observability. Therefore, the amount of mutual information gained per step can be a useful intrinsic reward to encourage informative actions. Moreover, we train the agent to predict the state features from past histories as an auxiliary task to help the agent focus on memorizing enlightening past events, i.e., previous actions or observations that have gained more information about the environment state.

We propose a two-stage approach to turn intuition into a practical and efficient method. The first stage extracts task-relevant but non-overlapping state and observation features by performing supervised learning using a state-contained dataset. Specifically, we predict the reward and the dynamics in the feature space while minimizing the mutual information between state and observation features. We perform task learning in the second stage, which involves training a transformer-based sequence model to output a history summary with maximal mutual information with the learned state features

8th Conference on Robot Learning (CoRL 2024), Munich, Germany.

as possible. The goal is to reduce the memory burden of the sequence model by encouraging it to avoid memorizing readily available observation features. With a purposefully designed structure, intrinsic rewards, and the auxiliary task, our approach outperforms other strong baselines in experiments on six domains. Moreover, we carry out successful sim-to-real transfers to verify that the policies learned in simulation can perform well in three robot manipulation tasks.

## 2 Preliminaries

We first discuss the partially observable Markov decision process [10, 11] (POMDP) framework, methods that minimize/maximize the mutual information, and describe our problem statement.

### 2.1 Partially Observable Markov Decision Process

A POMDP is defined by a tuple $(\mathcal{S}, \mathcal{A}, \Omega, T, R, O)$, where $\mathcal{S}, \mathcal{A}$, and $\Omega$ are the state space, the action space, and the observation space, respectively. In a POMDP, the transition function $T(s, a, s')$ governs state changes, and the observation function $O(a, s', o)$ controls how observations are emitted. As the current observation only partly reflects the state, acting optimally in a POMDP with a high level of partial observability often requires information gathering and a memory-based policy that selects actions based on the entire action-observation history $h_t = (o_0, a_0, \ldots, a_{t-1}, o_t)$ [12].

### 2.2 Minimizing and Maximizing Mutual Information

Mutual Information (MI) measures the dependence between two random variables $x$ and $y$:

$$I(x; y) = \mathbb{E}_{p(x,y)} \left[ \log \frac{p(x, y)}{p(x)p(y)} \right], \tag{1}$$

where $p(x, y)$ is the joint distribution and $p(x), p(y)$ are the marginals.

**Minimizing MI:** Cheng et al. [13] proposed to minimize $I(x; y)$ by minimizing an upper bound named Contrastive Log-ratio Upper Bound (CLUB), which is formulated as:

$$I_{\text{CLUB}} := \mathbb{E}_{p(x,y)}[\log p(y|x)] - \mathbb{E}_{p(x)}\mathbb{E}_{p(y)}[\log p(y|x)] \geq I(x; y) \tag{2}$$

In practice, $p(y|x)$ is often unknown and approximated by a learnable variational distribution $q(y|x)$.

**Maximizing MI:** Instead of maximizing $I(x, y)$, Deep InfoMax (DIM) [14] is a method that maximizes the MI between $x$ and a high-level representation $z^x$ of $x$. The process involves an encoder $E : \mathcal{X} \to \mathcal{Z}^x$ and a classifier $\sigma : \mathcal{X} \times \mathcal{Z}^x \to \mathbb{R}$ to discriminate between samples from $p(x, z^x)$ and the product of marginals $p(x)p(z^x)$. DIM uses the Jensen-Shannon MI estimator [15]:

$$I_{\text{DIM}} := \mathbb{E}_{p(x,E(x))}[-\text{sp}(-\sigma(x, E(x)))] - \mathbb{E}_{p(x)p(E(x))}[\text{sp}(\sigma(x, E(x)))] \leq I(x; E(x)), \tag{3}$$

where $\text{sp}(z) = \log(1 + e^z)$ is the softplus function. Maximizing $I(x, z^x)$ involves continuously updating $E$ and $\sigma$ by maximizing the estimator $I_{\text{DIM}}$ through gradient ascent.

In practice, we optimize Eq. (2) and Eq. (3) through samples; see the Appendix for more details.

### 2.3 Problem Statement

We focus on the asymmetric RL setting [3, 9, 6, 8] in which, *during training*, *both* the state $s_t$ and the observation $o_t$ are available while *only* $o_t$ is accessible *during execution*. Unlike an MDP, the goal here is to find a history-based (not reactive) policy $\pi(a_t|h_t)$ which maximizes the expected discounted return $J = \mathbb{E}\left[\sum_{t=0}^{\infty} \gamma^t R(s_t, a_t)\right]$, where $\gamma \in [0, 1)$ is a discounting factor. We use an information-based approach to leverage states during training for efficiently learning $\pi(a_t|h_t)$.

## 3 Method

Our method has two stages. First, we learn non-overlapping state and observation features from a fixed dataset containing privileged state information. The dataset is generated using a random agent

or a simple hard-coded planner. Next, we use the learned features to train a memory-based agent with a two-branch architecture using information-based intrinsic rewards and an auxiliary task.

## 3.1 Learning Task-Relevant and Non-Overlapping State and Observation Features

This stage aims to learn state features $z^s$ (encoded by a state encoder $\phi$) and observation features $z^o$ (encoded by an observation encoder $\psi$) with two desired properties. (**P1**): Firstly, we want the concatenated feature $z^s \oplus z^o$ to encode task-relevant state features. In other words, $z^s \oplus z^o$ is a compact "state", which is useful when the raw state is high-dimensional or contains task-irrelevant features. (**P2**): Secondly, we want $z^s$ to contain *unobservable* task-relevant features, while $z^o$ contains *observable* task-relevant features. With P1 and P2 satisfied, $z^s$ will encode only task-relevant but unobservable features inferred from the history as $z^o$ already contains features extracted from the observation. We can also view $z^s$ as the outcome of an ideal information-gathering process, which results in task-relevant hidden information not captured by the current observation (i.e., $z^o$).

We satisfy (P1) by training $z^s \oplus z^o$ given an action $a$ to be predictive of $z^{s'}$, $z^{o'}$ and the reward $r$, similar to bi-simulation studies for MDPs [16, 17, 18]. For (P2), Wang et al. [8] minimized the KL divergence between $\phi(s)$ and a fixed standard normal distribution by giving a penalty whenever features are derived from the $s$ instead of $o$. However, we will empirically show (in Section 5.2.1) that this approach does not satisfy (P2). Instead, inspired by [19], we directly minimize the mutual information $I(z^s; z^o)$ using Eq. (2). However, minimizing only $I(z^s; z^o)$ may result in an undesired scenario, where $I(z^s; z^o)$ is small but only $\phi(s)$ or $\psi(o)$ captures all good features, and the remaining contains no useful information. Therefore, we further regularize learning with maximizing $I(o; z^o)$ and $I(s; z^s)$ using Eq. (3). Algorithm 1 summarizes this stage (more details in the Appendix C).

---

**Algorithm 1** Learning $z^s$ and $z^o$ (more details in Appendix C)

---

**Require:** Dataset $\mathcal{D} = \{(s_i, o_i, a_i, r_i, s'_i, o'_i)\}_{i=1}^N$
1: **repeat**
2:     Sample a batch of data $\{(s_i, o_i, a_i, r_i, s'_i, o'_i)\}_{i=1}^B \sim \mathcal{D}$
3:     Compute current features $z_i^s = \phi(s_i)$, $z_i^o = \psi(o_i)$
4:     Compute reward and next features using a dynamics model $g$: $\hat{r}_i, \hat{z}_i^{s'}, \hat{z}_i^{o'} = g(z_i^s, z_i^o, a_i)$
5:     Compute target features $z_i^{s'} = \phi(s'_i)$, $z_i^{o'} = \psi(o'_i)$
6:     Compute $\mathcal{L}_r = (r_i - \hat{r}_i)^2$, $\mathcal{L}_s = \|\text{stop-grad}(z_i^{s'}) - \hat{z}_i^{s'}\|_2^2$, $\mathcal{L}_o = \|\text{stop-grad}(z_i^{o'}) - \hat{z}_i^{o'}\|_2^2$
7:     Compute $\mathcal{L}_{\text{CLUB}}$ using Eq. (2) to minimize $I(z^s; z^o)$
8:     Compute $\mathcal{L}_{\text{DIM}}$ using Eq. (3) to maximize $I(o; z^o)$ and $I(s; z^s)$
9:     Update $\phi, \psi, g$ to minimize $\lambda_r \mathcal{L}_r + \lambda_s \mathcal{L}_s + \lambda_o \mathcal{L}_o + \lambda_{\text{CLUB}} \mathcal{L}_{\text{CLUB}} + \lambda_{\text{DIM}} \mathcal{L}_{\text{DIM}}$
10: **until** convergence

---

## 3.2 Learning Partially Observable Tasks

After learning $z^s$ and $z^o$, we now introduce a memory-based agent with two-branch architecture (see Fig. 1) for task learning using RL. The *memory-based* top branch is responsible for encoding the history using a GPT-v2-based sequence model [20, 21, 22]. In contrast, the *memoryless* bottom branch encodes the current observation. When the agent finishes gathering information, the top branch will ideally output features in $z^s$, which are not captured in $z^o$. For the bottom branch, we directly use $\psi(o)$ to initialize it. A two-branch architecture is not new by itself (see [23, 24, 25, 26]) and has been found to offer no performance gain over the single-branch (no bottom branch) architecture [23]. Unlike the previous work that used this architecture, by having non-overlapping $z^s$ and $z^o$ and using $z^s$ to bias the top branch's learning, we alleviate the top branch's memory burden by training it to focus only on features not readily observed.

Because $z^s$ is the ideal outcome of the information-gathering process, we encourage information-gathering by rewarding actions that produce $h_t$ that can maximize $I(h_t; z_t^s)$. Moreover, we use maximizing $I(h_t; z_t^s)$ as an auxiliary task to help memorize important information from history.

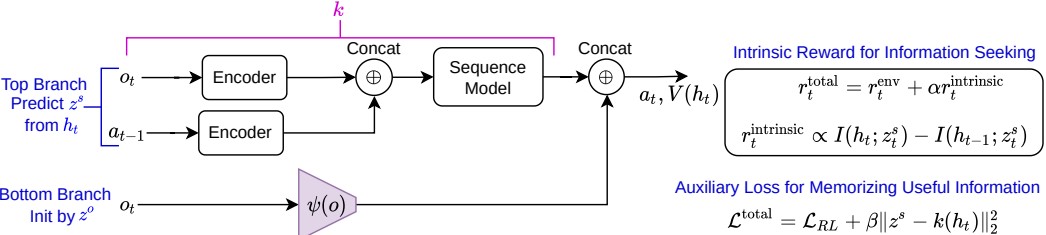

Figure 1: Our proposed agent structure with two branches ($\oplus$ denotes vector concatenation). The observation encoder $\psi(o)$ is directly used for the bottom branch. An intrinsic reward calculated based on $I(h_t; z^s)$ encourages information seeking. Moreover, to better memorize useful information, besides being trained to minimize the RL losses, the top branch ($k$) is trained to predict the state feature $z^s$ from the history $h_t$.

**Encouraging Information Seeking.** By maximizing $I(h_t; z_t^s)$, the agent is encouraged to gather information by acting to get $h_t$ that encodes maximal information with $z_t^s$. We closely follow [27] and maximize the mutual information by maximizing the following lower bound [28]:

$$I(h_t; z_t^s) = H(z_t^s) - H(z_t^s|h_t) \geq H(z_t^s) + \mathbb{E}_{h_t \sim \pi}[\log f_\theta(z_t^s|h_t)]$$

$$= H(z_t^s) + \log f_\theta(z_t^s|h_0) + \mathbb{E}_{h_t \sim \pi}\left[\sum_{t'=0}^{t-1} \log \frac{f_\theta(z_t^s|h_{t'+1})}{f_\theta(z_t^s|h_{t'})}\right], \tag{4}$$

where $f_\theta(z^s|h)$ serves as an approximation to the true conditional distribution $p(z^s|h)$. Similar to [27], we assume $f_\theta(z^s|h)$ follows a Gaussian distribution centered around a deterministic history embedding $f_\theta(h)$ with variance $\rho^2 I$. We parameterize $f_\theta(h)$ using the GPT-based sequence model and train $f_\theta(h)$ to predict $z^s$ across batches of $(h, z^s)$ samples by minimizing:

$$\mathcal{L}_f = \mathbb{E}_{(h,z^s) \sim \pi}\left[\|z^s - f_\theta(h)\|_2^2\right] \tag{5}$$

Since only the third term of Eq. (4) depends on $\pi$, we interpret $\log \frac{f_\theta(z_t^s|h_t)}{f_\theta(z_t^s|h_{t-1})}$ as an intrinsic reward term $r_t^{\text{intrinsic}}$. This intrinsic reward captures per-step incremental information gain that the current history $h_t$ reveals about $z_t^s$ compared to the previous $h_{t-1}$. With a weighting factor $\alpha$, our final reward therefore becomes $r_t^{\text{total}} := r_t^{\text{env}} + \alpha r_t^{\text{intrinsic}}$. In practice, we calculate the intrinsic rewards as $r_t^{\text{intrinsic}} = d_{\text{cosine}}(z_t^s, f_\theta(h_t)) - d_{\text{cosine}}(z_t^s, f_\theta(h_{t-1}))$, with $d_{\text{cosine}}$ being the cosine similarity distance.

**Encouraging Information Retention.** While the intrinsic reward encourages information gathering, we also want the agent to memorize important observations (or actions). For this purpose, we train the agent's top branch ($k$ in Fig. 1) to predict $z^s$ over a batch of episodes as an auxiliary task using the same Eq. (5) but replacing $f$ with $k$. While $f$ and $k$ seem similar, the key difference is that $k$ is part of the agent (it is the top branch) while $f$ is not ($f$ is a network that approximates $p(z^s|h)$). Training the agent this way can be considered an indirect and "online" way to maximize $I(h; z^s)$ so that the agent can mentally memorize important events within the history $h$ that increased $I(h; z^s)$.

## 4  Related Work

Leveraging privileged information during policy learning has been successful in various settings, most notably in the multi-agent setting. Usually, joint quantities (e.g., observations, histories [29], or actions) from all agents, the state [30], or their combinations [31, 32, 33, 34], are used to train a *centralized* critic to aid training *decentralized* actors. In the single-agent setting, Pinto et al. [3] leveraged the state availability in simulation to improve an image-based policy. Under partial observability, Belief-grounded Networks [5] use an auxiliary loss to reconstruct the privileged state distribution information from history representations. Using a value-based method, Baisero et al. [6] uses the state access to learn a history-based critic. Using the actor-critic framework, Baisero and Amato [9] learn a history-state-based critic to help learn a history-based actor. Concurrent with our work is [35], which also proposes informational rewards to encourage information gathering.

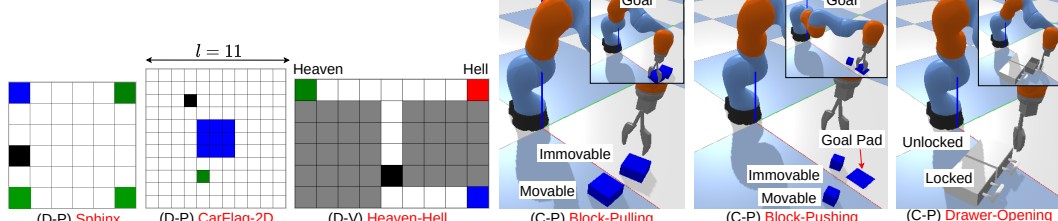

Figure 2: Domains. D/C: **d**iscrete/**c**ontinuous actions. V/P: **v**ector-based/**p**ixel-based observations.

While these methods use the privileged information as additional input (e.g., to train a critic), we first find a compact and task-relevant representation of the privileged information (similar to [8]). However, we diverge from [8] by directly using privileged information for encouraging information gathering. Our information-based approach shares similarities with a meta-learning work [27], which uses the mutual information between the history and the training task ID to compute intrinsic rewards to encourage information gathering. While [27] learns two policies for exploration (with the intrinsic rewards) and exploitation (with the environment rewards), we learn a single policy using a combined reward and add an auxiliary loss to improve memorization. Our idea of minimizing the overlapping between $z^s$ and $z^o$ has connections with the mixed-observability framework [36], which removed the overlapping between state and observation to reduce the computation for state distribution tracking. However, the proposed method [36] requires knowing the complete dynamics (i.e., planning-based) and the state and observation structure, while ours assumes no such knowledge.

## 5 Simulation Experiments

We perform experiments on three grid-world domains with discrete action spaces and three robot manipulation domains with continuous action spaces (see more details in Appendix B).

### 5.0.1 Discrete Action Space Domains

In Sphinx [8], CarFlag-2D [37], and Heaven-Hell [6] (see Fig. 2), an agent (■) must reach a *randomly generated* goal position (■). Specifically, the goal can be one of the green corners in Sphinx, left/right ends in Heaven-Hell, and any cell in CarFlag-2D. Normally, the observation only contains the agent's location. Only when entering the information region(■), the agent can additionally observe the goal. For instance, when entering the blue cell in Heaven-Hell, the side of the goal (left/right) is observed. Similarly, in the Sphinx and CarFlag-2D, the observation will encode the goal cell if entering the blue region. Rewards are given sparsely: +1 when the agent reaches the goal, −1 when it reaches the trap locations (■) if any, and 0 otherwise. In Sphinx, an additional penalty of −0.2 is given when entering the blue cell, making the information-gathering process costly. In these domains, the state encodes both the current observation and the goal cell.

### 5.0.2 Continuous Action Space Domains

In Block-Pulling, Block-Pushing, and Drawer-Opening (modified from domains in [38]), a robot must pull, push, or open the only *movable* object (a block or a drawer) among two, which appear the same under a top-down depth camera. The challenge is that the mobility of the objects is hidden if only relying on the current depth image (the observation). An optimal agent must perform information-gathering actions to determine the objects' mobility and then memorize the information while manipulating the movable object to achieve the tasks. Here, the state is also pixel-based, created by concatenating the observation and a 1-channel image that masks everything except the movable object. The reward is +1 only when finishing the tasks. To overcome the reward sparsity, we give 80 episodes of demonstrations to pre-populate the replay buffers of *all agents*. Moreover, because these tasks exhibit spatial symmetry [39], we use random SO(2) rotations (see Appendix F), applied consistently for all observations and actions within an episode, for data augmentation.

## 5.1 Baselines

We use DDQN [40] (discrete action spaces) and SAC [41] (continuous action spaces) as the base RL algorithms for our agent and all baselines. We make them memory-based using a simplified version of GPT-v2 [20, 22] as the sequence model. All following baselines use the two-branch architecture like ours (see Appendix A).

**DTQN, TSAC**: These variants [21, 22] are based on DQN and SAC and use a **t**ransformer (a simplified version of GPT-v2 [20]) as the sequence model. **ZP-DRQN, ZP-RSAC**: Ni et al. [42] proposed the next hidden state prediction as an unsupervised auxiliary task to improve learning under partial observability. We applied their method to DRQN and RSAC and used a GRU [43] instead of a transformer here to satisfy the recurrent encoder property in [42] (i.e., a transformer is *not* a recurrent model). **BA-DTQN, BA-TSAC**: The **b**iased **a**symmetric version of DTQN and TSAC. For BA-DTQN, $Q(s, a)$ and $Q(h, a)$ are learned during training, and $\arg\max_a Q(h, a)$ is the selected action during deployment. For BA-TSAC, only the state-based critic $Q(s, a)$ is learned to help learn a history-based actor. These asymmetric agents can sometimes perform well [3] but were found [6, 9] to introduce learning bias, i.e., $Q(s, a)$ is a biased estimate of $Q(h, a)$. **UA-DTQN, UA-TSAC**: We implement **u**nbiased **a**symmetric variants of DTQN and TSAC using the framework proposed in [6, 9] (originally applied to DRQN [44] and A2C [45]). To fix the bias issue, during training, these methods learn $Q(h, s, a)$ (instead of $Q(s, a)$) and $Q(h, a)$ and use $Q(h, a)$ to pick actions during execution. **B-DQN**, **B-SAC**: **B**eliever [8] originally trains a GRU-based history encoder, similar to our first stage. For a fair comparison, we use the same GPT to replace the GRU and the same data set for our agent and these agents. Because the encoders already encode the history, their outputs are directly used for memoryless RL algorithms, i.e., DQN or SAC in our case. Furthermore, we fine-tune the history encoders to strengthen these baselines, following the public code.

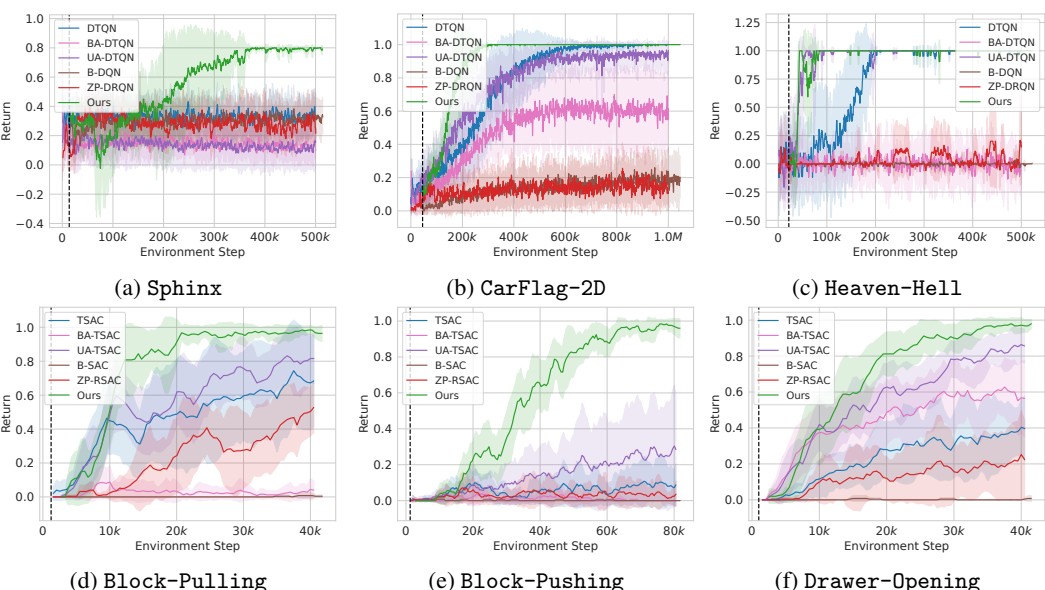

(a) Sphinx      (b) CarFlag-2D      (c) Heaven-Hell

(d) Block-Pulling      (e) Block-Pushing      (f) Drawer-Opening

Figure 3: Learning curves averaged over five random seeds with shaded mean and one std. Dashed vertical lines denote when the task learning stage of our method and B-DQN/B-SAC starts.

## 5.2 Experimental Results

As shown in Fig. 3, our method is generally superior to all baselines in all domains. Among baselines with state access like ours, only unbiased asymmetric methods (UA-DTQN/-TSAC) can compete with ours in Heaven-Hell, Block-Pulling, and Drawer-Opening. Most surprisingly, Believer-based baselines (B-DQN/-SAC) do not perform well in all domains despite our significant effort to strengthen them (replacing GRU with GPT, hyper-parameter tuning, and online fine-tuning). When information seeking is hard (e.g., in Sphinx with a clarifying penalty, or in Heaven-Hell with

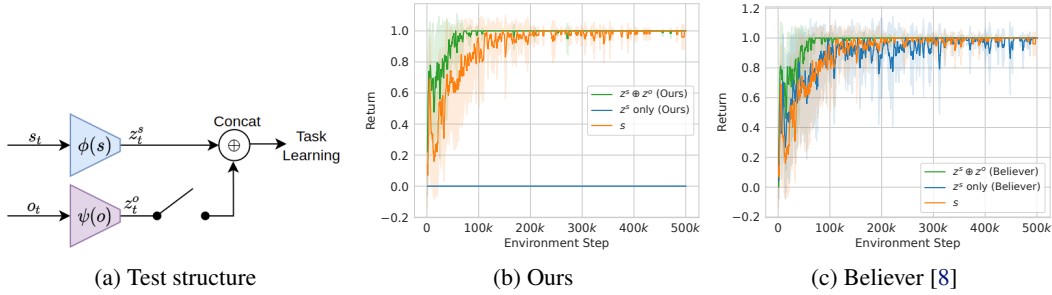

(a) Test structure          (b) Ours          (c) Believer [8]

Figure 4: a) Overlapping test: using $z^s$ or concatenating $z^s$ and $z^o$ for task learning; b-c) Learning curves in `Heaven-Hell` when using our method (Section 3.1) or Believer [8] to learn $z^s$ and $z^o$.

a goal-avoided path for goal information), only our agent can perform well by using the intrinsic rewards. When the information seeking is easier, our method still outperforms (e.g., in `CarFlag-2D`, `Block-Pulling`, and `Drawer-Opening`) thanks to pre-trained compact features.

### 5.2.1 Learned High Quality and Non-overlapping Representations

To show that our method can achieve both (P1) and (P2) in Section 3.1, we perform task learning in `Heaven-Hell` using **a)** $z^s \oplus z^o$, **b)** only $z^s$, and **c)** raw state $s$. We compare two ways to learn the representation: our method and Believer [8]. From Fig. 4, $z^s \oplus z^o$ yields better performance than the raw state. This indicates that our method can learn high-quality state features. As expected, using $z^s$ alone fails to solve the task because $z^s$ will only contain the features of the goal's position. The task can only be solved with the aid of $z^o$, which contains the agent's position. In contrast, when using Believer [8] to learn $z^s$ and $z^o$, using either $z^s \oplus z^o$ or $z^s$ makes no difference: both can solve the task, indicating that $z^s$ also contains the agent's position, which is already contained in $z^o$.

### 5.2.2 Ablation Studies

In this section, we perform ablation studies to analyze our key design choices: **a)** using GPT instead of a recurrent neural network (e.g., a GRU) as the sequence model and **b)** using both the auxiliary losses and the intrinsic rewards. We only report representative results in this section; see the Appendix D for more details.

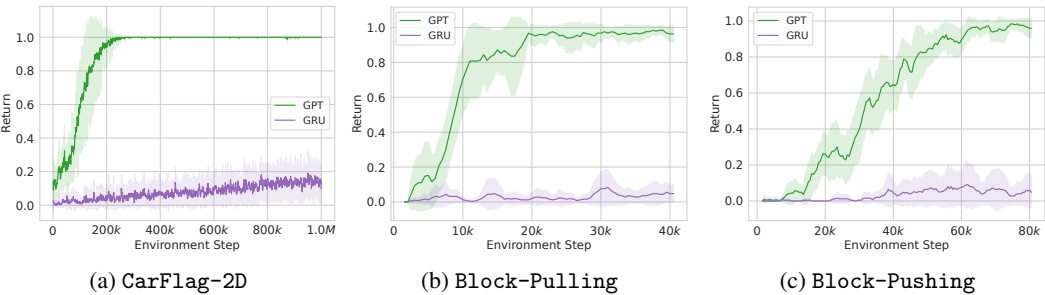

(a) `CarFlag-2D`          (b) `Block-Pulling`          (c) `Block-Pushing`

Figure 5: Comparing when using GPT or a GRU as the sequence model in our approach.

**Using GPT instead of a recurrent neural network.** While a GRU [43] might have trouble memorizing important information in long sequences (i.e., saw important observations but forgot), a GPT can attend to and memorize any past information. This property is crucial for our agent to maximize $I(h; \phi(s))$ effectively. From the performance gap in Fig. 5, we can see the important role of GPT.

**Using both auxiliary losses and intrinsic rewards.** While the auxiliary losses help information retention, intrinsic rewards are needed to encourage information gathering directly. As shown in Fig. 6, in domains when information gathering is not challenging, such as in `CarFlag-2D` (the information region is quite large) and the robot domains (agents are provided with demonstration episodes), using auxiliary loss is sufficient for good performance. In `Sphinx`, when information is harder to seek because of the involved cost, the intrinsic rewards are essential to improve the performance.

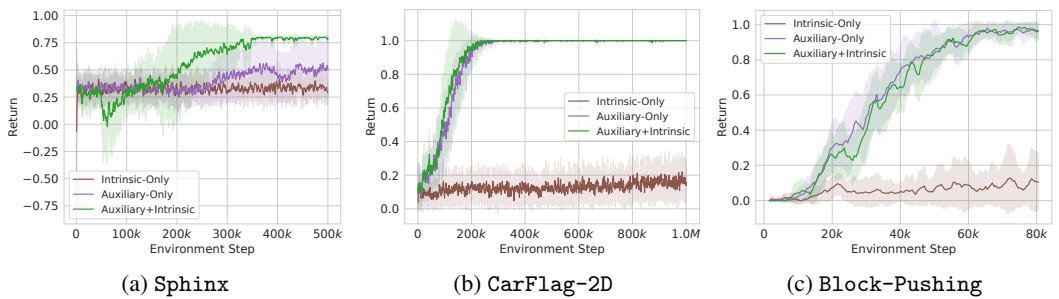

| (a) `Sphinx` | (b) `CarFlag-2D` | (c) `Block-Pushing` |

Figure 6: Comparing using intrinsic rewards or auxiliary losses versus using both.

# 6 Sim-To-Real Transfers

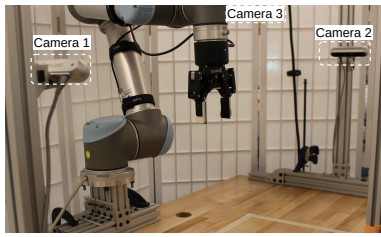

Figure 7: Real robot setup.

| Domain | Ours | UA-TSAC |
|---|---|---|
| `Block-Pulling` | 45/50 | 34/50 |
| `Block-Pushing` | 44/50 | 12/50 |
| `Drawer-Opening` | 41/50 | 32/50 |

Table 1: Average success rates of sim-to-real transfers between our method and UA-TSAC (best baseline).

To verify the performance of the learned policy in simulation, we perform sim-to-real transfers using the best policies in simulation to a UR5 robot (see Fig. 7). To best obtain non-occluded top-down depth images, we combine the point clouds from two side-view cameras and one top-down camera and project at the gripper's position (see Appendix E). Furthermore, during training, we add Perlin noise [46] to the observation to reduce the gap between the simulated and the real-world depth images. We perform 50 evaluation episodes, divided equally into two cases when the policies first manipulate the immovable or movable objects. Results from Table 1 show that the learned policies can be zero-shot transferred in the real world (see our supplementary video for more details), and our method performs better than the best baseline (UA-TSAC). During sim-to-real transfers, the main failure mode is when the robot goes down to perform actions; collisions with objects (e.g., blocks, drawers) might happen. This can trigger the robot's protective stop and terminate the episode. Additionally, in `Drawer-Opening`, the transferred policies sometimes clumsily move one drawer far away from the other, creating a novel scene never seen in simulation.

# 7 Conclusion and Limitations

**Conclusion.** This work proposed an information-based method that directly uses the mutual information between the state and history to encourage information-gathering under partial observability. We also tackle memorization by incorporating an auxiliary task to predict the state feature from the history to alleviate the agent's memory burden by only requiring its memory component to memorize unobservable task-relevant features. Experiments show that our method is effective and scalable to domains with high-dimensional states and observations with deployable working policies.

**Limitations.** Our method's biggest limitation is the challenge of learning good and non-overlapping representations at the first stage. In practice, we need to select the right dataset to train with. We can use random interactions in the grid-world domains, but expert demonstrations are needed for training in the robot domains so that the representations are useful. In practice, to guarantee the learned representations can satisfy both (P1) and (P2), we need to iterate between hyper-parameter tuning and performing the overlapping test in Section 5.2.1. We also want to note that while the second stage works optimally with high-quality representations, our method remains effective with less ideal features, as the agent can still refine these through reinforcement learning.

**Acknowledgments**

This research is supported by grants from the Army Research Office (W911NF20-1-0265), NASA (80NSSC19K1474), and the National Science Foundation (NSF 2024790, 1750649, 2107256, 2314182, 2409351).

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

# A  Details of Agents

## A.1  DTQN, TSAC

These are variants of DQN and SAC, made memory-based by using a transformer as the sequence model as shown in Fig. 8 and Fig. 9. Similar models have been explored in previous work [21, 22].

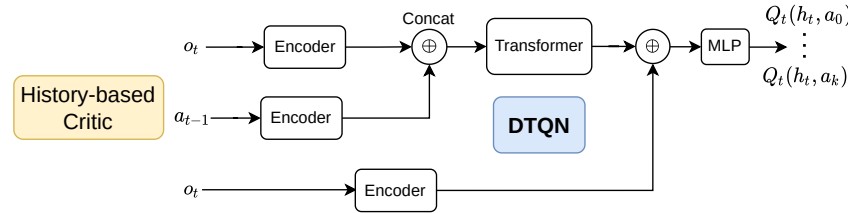

Figure 8: Architecture of DTQN.

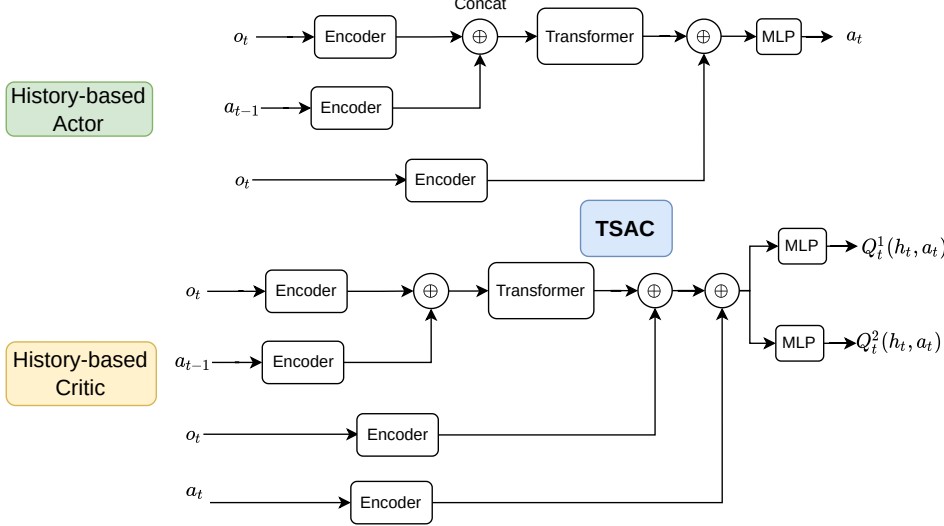

Figure 9: Architecture of TSAC.

## A.2  ZP-DRQN, ZP-RSAC

These agents [42] are similar to DTQN and TSAC, except they use a recurrent sequence model instead of a transformer. Importantly, using a *recurrent* sequence model (e.g., a GRU [43]) is required (see [42]). Additionally, these agents are regularized with a self-predictive auxiliary task of predicting the next latent state $z$ from a history $h$. Specifically, given a recurrent encoder $f_\phi : \mathcal{H} \to \mathcal{Z}$ and a latent dynamics model $g_\theta : \mathcal{Z} \times \mathcal{A} \to \mathcal{Z}$, the auxiliary task is to minimize:

$$\mathcal{L}_{\mathrm{aux}} = \| g_\theta(f_\phi(h), a) - f_{\bar\phi}(h') \|_2^2 , \tag{6}$$

where $\bar\phi$ is the target network of $\phi$.

## A.3  BA-DTQN, BA-TSAC

In BA-DTQN [6] (see Fig. 10), a state-based critic $Q(s, a)$ and a history-based critic $Q(h, a)$ are learned to leverage the state availability during training but not during execution (i.e., no state is available therefore we cannot use $Q(s, a)$ during execution). Unfortunately, as pointed out in [6],

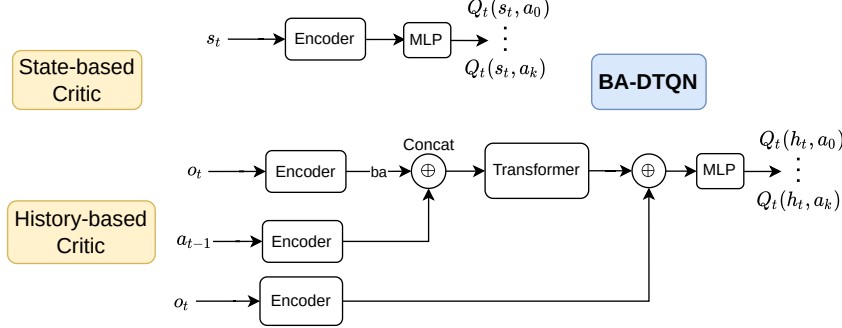

Figure 10: Architecture of BA-DTQN.

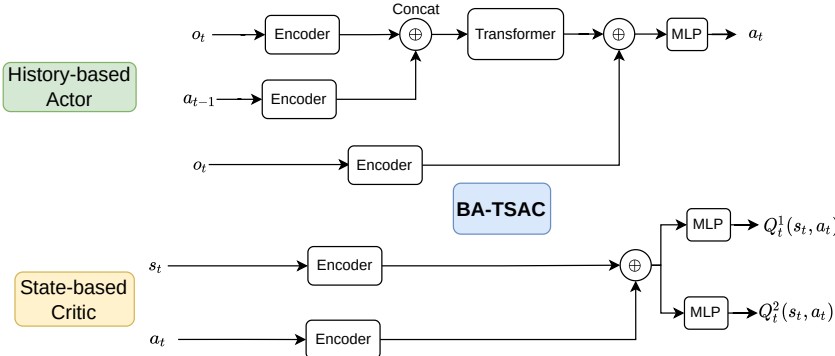

Figure 11: Architecture of BA-TSAC.

$Q(s,a)$ is not mathematically well-defined and is generally a biased estimate of $Q(h,a)$, which is used to select actions during execution.

The difference between BA-TSAC (see Fig. 11) and TSAC is that the critic is trained additionally using state input during training. Specifically, we learn a state-based critic $Q(s,a)$ instead of the history-based $Q(h,a)$. Similar to BA-DTQN, BA-TSAC also has bias. For BA-TSAC, during execution, actions are computed using a history-based actor.

### A.4  UA-DTQN, UA-TSAC

In UA-DTQN [6] (see Fig. 12), a history-state-based critic $Q(h,s,a)$ and a history-based critic $Q(h,a)$ are learned. Unlike BA-DTQN with $Q(s,a)$, $Q(s,h,a)$ can be well-defined and has been proven to be an unbiased estimate of $Q(h,a)$. During execution, actions are selected using $Q(h,a)$.

Unlike BA-TSAC, UA-TSAC [9] (see Fig. 13) combines *both* state and history features to train the critic, i.e., we learn a history-state-based critic $Q(s,h,a)$. Similar to UA-DTQN, UA-TSAC does not introduce learning bias. During execution, actions are computed from a history-based policy.

### A.5  B-DQN, B-SAC

The architectures of these agents are depicted in Fig. 14. These agents are based on Believer [8], which leveraged the state availability to train an agent in three stages:

**Stage 1.** Learning compact state representations with state-labeled transitions, i.e., a batch of samples $(s,o,a,r,s',o')$. This stage is similar to our first stage (see Algorithm 1) but without the information-based regularizations. Instead, the authors proposed to regularize the KL divergence $\text{KL}[\phi(s)\|\mathcal{N}(0,1)]$ to avoid overlapping features between $\phi(s)$ and $\psi(o)$ by giving penalty when-

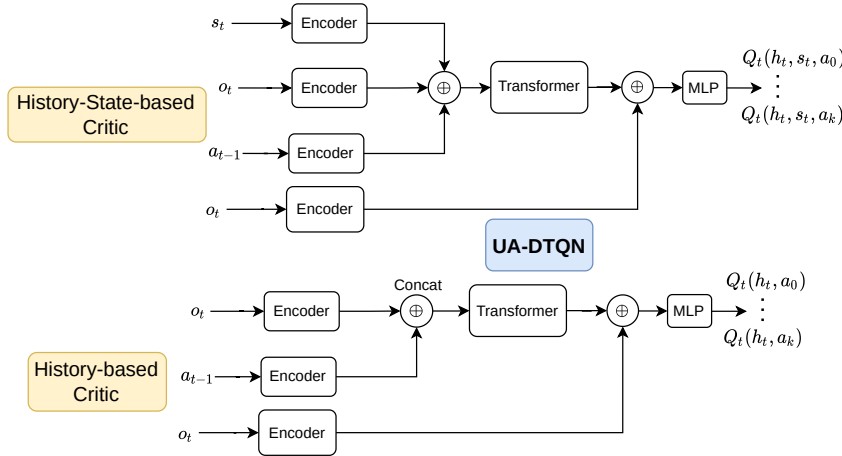

Figure 12: Architecture of UA-DTQN.

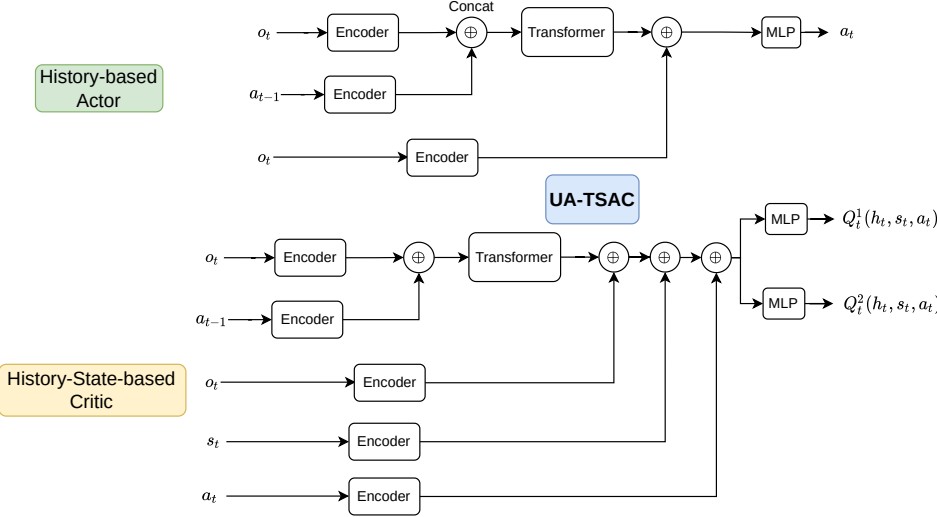

Figure 13: Architecture of UA-TSAC.

ever $\phi(s)$ is used to derive features. This, however, does not avoid the overlapping issue between learned state and observation features, as shown in our experiment (see Fig. 4).

**Stage 2.** Learn a recurrent history model $p(\phi(s)|h)$ with variational autoencoders [47] (VAE) by maximizing the joint log-likelihood $p(\phi(s), h))$ averaged over $(s, h)$ samples.

**Stage 3.** Use the history module $p(\phi(s)|h)$ for task learning. First, samples are drawn from the VAE to derive a history summary. Then, this summary is used as the "states" for task learning using memoryless RL algorithms. The authors optionally fine-tune $p(\phi(s)|h)$ with the on-policy data.

As the original paper applied their method for PPO [48], which is on-policy, we had to modify the method to apply to DQN and SAC, resulting in B-DQN and B-SAC. In Stage 2, to fairly compare with other baselines, we replace GRU in the history model with the GPT model used in other baselines. Moreover, in Stage 3, we fine-tuned the history module for every domain (as used in the original code). Finally, the sequence model of B-SAC is shared between the actor and the critic, following the original code.

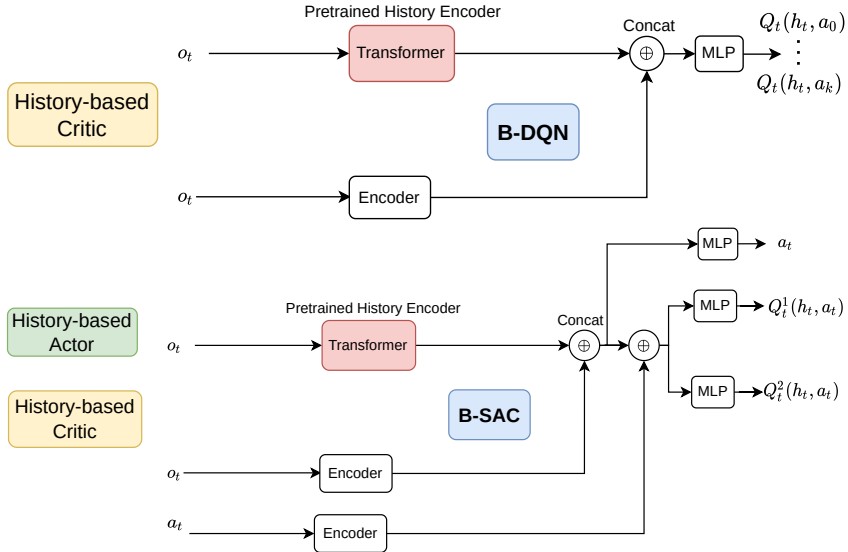

Figure 14: Architectures of B-DQN and B-SAC with a pre-trained history encoder from Believer [8]. We change the history encoder from GRU-based to transformer-based for a fair comparison with other agents. For B-SAC, we use a shared history module, similar to the original code.

## A.6 Hyper-parameters

For DDQN [40], we use an epsilon-greedy exploration strategy with a linear schedule, starting at $\epsilon = 1.0$ and ending at $\epsilon = \frac{1}{T}$ with $T$ being the episode length. The schedule time is equal to 10% of the total training timesteps. We use a batch size of 64 episodes. For continuous actions, we use SAC [41]. We automatically tune the entropy temperature, initializing at $0.01$. The chosen target entropy is equal to the negation of the action dimension. We use the discount factor $\gamma = 0.99$. We use a batch size of 64 episodes. Other hyper-parameters are in Table 2 with shared parameters and ones specific for each agent.

Table 2: Hyper-parameters used for RL agents. HH: `Heaven-Hell`, S: `Sphinx`, CF: `CarFlag-2D`.

| Agent | Hyper-parameter | Value |
|---|---|---|
| Shared among all agents | Episode Length | 50 |
| | Discount Factor | 0.99 |
| | Replay Buffer Size | 1M: grid world domains, 100k: robot domains |
| | Target Update Rate | 0.005 |
| | Actor Learning Rate | 3e-4 |
| | Critic Learning Rate | 3e-5: CF and S; 3e-4: other |
| | Batch size | 64 |
| SAC | Initial Entropy Temperature | 0.01 |
| | Update Per Step | 0.25: discrete domains, 1.0: robot domains |
| ZP-DRQN | Loss weighting | 1.0: discrete domains, 0.1: robot domains |
| Ours | Loss Weighting | 0.5 for all domains |
| | Reward Weighting | 10.0: HH, S, 0.1: robot domains, 0.0: CF |
| B-DQN, B-SAC | Latent Dimension | 32 |
| | X-Dim | 16 |
| | Z-Dim | 16 |

## A.7 Network Structures

**FC**($n$): a fully connected layer with $n$ outputs; **Conv**($f, s$): a convolutional layer with filter size $f \times f$ and stride $s$; **R**: the ReLU activation function; **MaxPool**($w$): a max pooling layer with window size $w$; **T**(H, N, HS, D): Transformer with H heads, N layers, hidden size HS, and the dropout rate D; **GRU**(N, HS): GRU with N layers and hidden size HS.

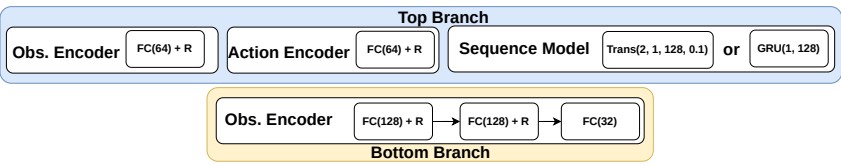

Figure 15: Network structures used in `Heaven-Hell`.

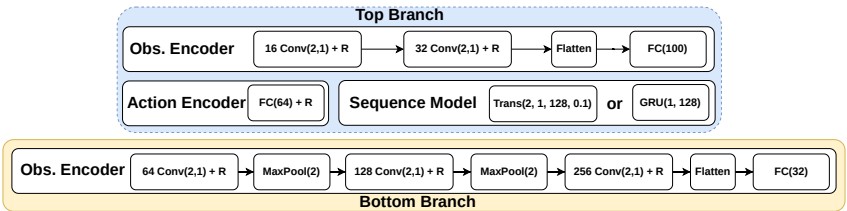

Figure 16: Network structures used in `CarFlag-2D`.

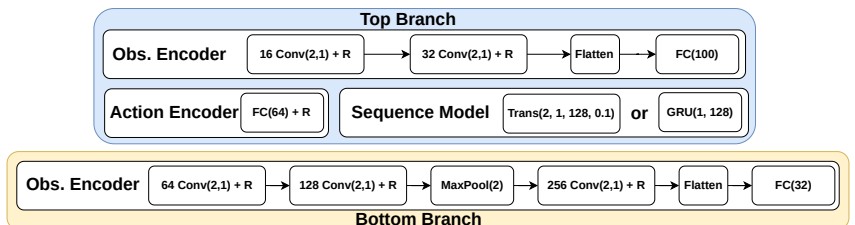

Figure 17: Network structures used in `Sphinx`.

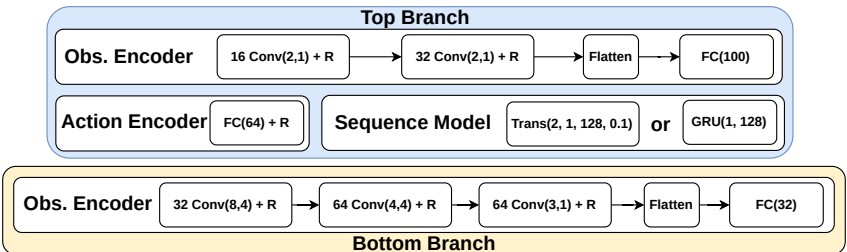

Figure 18: Network structures used in robot domains.

# B Details of Domains

## B.1 Sphinx

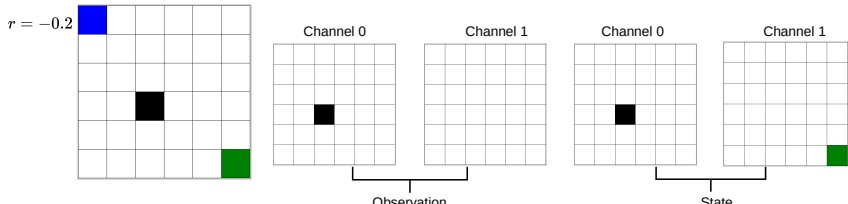

Figure 19: `Sphinx` domain with two-channeled pixel-based observations and states. Channel 1 of the observation reveals the goal cell (green) only when the agent enters the blue cell. In contrast, the same channel of the state always reveals the goal cell regardless of the agent's position.

In this domain (see Fig. 19), an agent must visit the goal cell, which can be in one of three corners except the top-left one. The agent must visit the information cell (blue) at the top-left corner to know the current corner of the goal. However, there is a cost when going to the information cell.

**Action.** Move-Right, Move-Left, Move-Up, Move-Down

**Observation.** A $6 \times 6 \times 2$ image with the first channel encodes the agent's position and the second encodes the goal's position. The second channel only contains the goal information when the agent enters the blue cell.

**State.** A state has the same structure as an observation, but the second channel always contains the goal information.

**Reward.** $+1$ when reaching the goal, $-0.2$ when visiting the information cell, and $0$ otherwise.

## B.2 CarFlag-2D

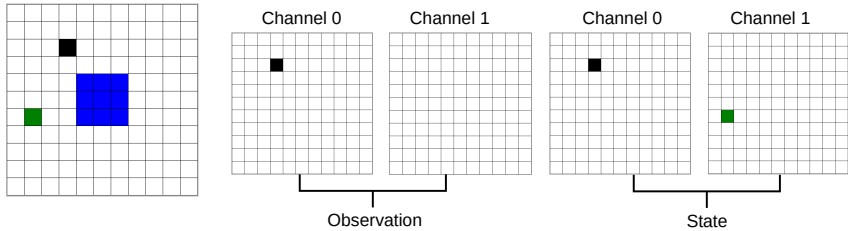

Figure 20: `CarFlag-2D` domain with two-channeled pixel-based observations and states. Channel 1 of the observation reveals the goal cell only when the agent enters the blue region. In contrast, the same channel of the state always reveals the goal cell.

In this domain (see Fig. 20), an agent must visit the goal cell (green) to finish the task. The goal cell, however, is only present in the observation when the agent visits the information region (blue).

**Action.** Move-Right, Move-Left, Move-Up, Move-Down

**Observation.** A $11 \times 11 \times 2$ image with the first channel encodes the agent's position, and the second encodes the goal's position. The second channel only contains the goal information when the agent enters the blue region.

**State.** A state has the same structure as an observation, but the second channel always contains the goal information.

**Reward.** $+1$ when reaching the goal, and $0$ otherwise.

## B.3  Heaven-Hell

In this domain, an agent must visit heaven (green cell) to finish the task. The goal cell can be either on the left or on the right side with $50\%$ probability. To observe the side of the goal (left or right), the agent must visit the priest, who resides in the bottom right corner.

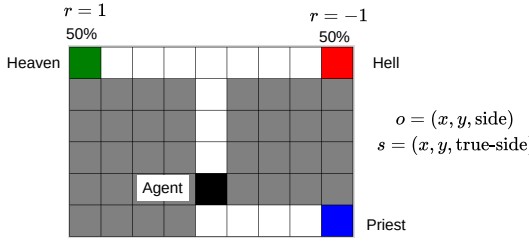

Figure 21: The `Heaven-Hell` domain with vector-based observations and states.

**Action.** Move-Right, Move-Left, Move-Up, Move-Down

**Observation.** A vector consists of the agent's position and the side information. The side information can take the value of $0$ (no information), $1$ (heaven on the right), or $-1$ (heaven on the left).

**State.** Like the observation, but the true side of the goal is always revealed.

**Reward.** $+1$ when reaching heaven, $-1$ when reaching hell, and $0$ otherwise.

## B.4  Robot Domains

In these domains, the agent must manipulate the only movable object among two objects, which are exactly the same under the top-down depth image observation.

**Action.** An action $a = (\delta_x, \delta_y, \delta_z, \delta_r)$, where $\delta_{xyz} \in [-0.05, 0.05]$ are the displacements of the gripper in the XYZ axes, and $\delta_r \in [-\pi/8, \pi/8]$ is the angular rotation around the Z axis.

**Observation.** All robot domains share the same observation: the top-down depth image taken from the camera centered at the gripper's position. Two fingers of the gripper are projected on the image.

**State.** The state also has two channels. The first channel is the first top-down depth image of the observation. While the second channel of the observation is non-informative, the second channel of the state is an image that masks everything except the movable object (see Fig. 22) in which the movable objects are colored red for visualizations).

**Reward.** In `Block-Pulling`, the agent receives a reward of 1.0 only when the two blocks are in contact. In `Block-Pushing`, the agent receives a reward of 1.0 only when the movable block is within 5 cm from the center of the goal pad. In `Drawer-Opening`, the agent receives a reward of 1.0 only when the unlocked drawer is opened more than 5 cm.

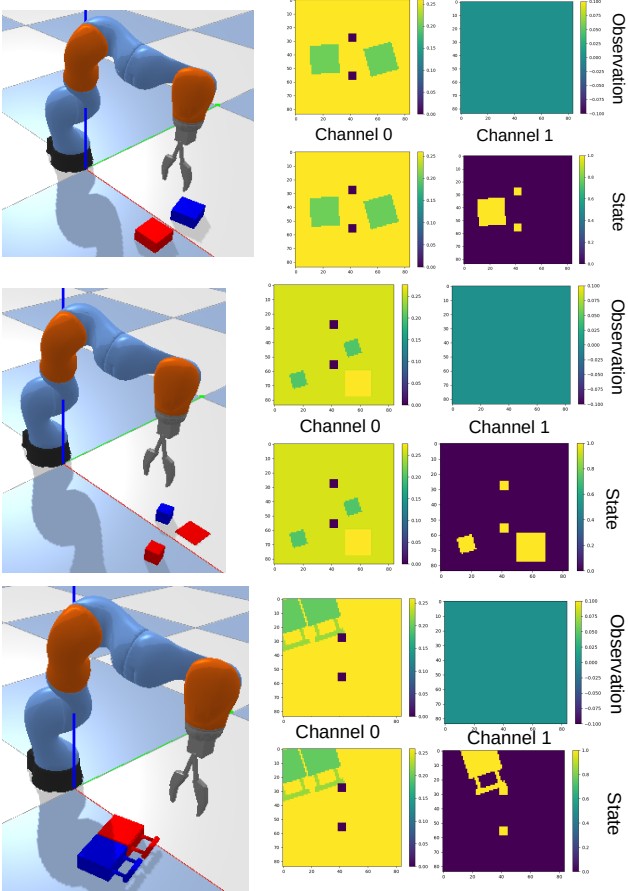

Figure 22: Visualization of an observation and a state in `Block-Pulling`, `Block-Pushing`, and `Drawer-Opening`. The movable object is the red one. The state and the observation have two channels, the first being the top-down depth image. In `Block-Pulling`, the second channel of the state reveals the movable object and the gripper. In `Block-Pushing`, the second channel reveals the movable block, the gripper, and the goal pad. In `Drawer-Opening`, the second channel in the state reveals the unlocked drawer and the gripper.

# C Representations Training Details

## C.1 Training Data Generation

`Heaven-Hell`, `CarFlag-2D`, `Sphinx`: In these domains, we use a uniform random agent to generate training samples, each is a transition $(s, o, a, r, s', o')$. For the number of samples used in each domain, please see Table 3.

In the robot domains, we use the same number of demonstrations (80 episodes) to learn the representations during task learning. Furthermore, we augment the training data using random rotations per transition as used in [49] (also see Appendix F). Finally, we describe the planners used to generate the demonstrations in these domains.

**Planner in `Block-Pulling`:** The planner randomly selects a block and attempts to pull it to the other block direction until the task is accomplished. If, for a while, the position of the selected block remains unchanged, the planner will move the gripper to the other block and repeat the pulling.

**Planner in `Block-Pushing`:** A block is randomly chosen and pushed toward the goal pad. If the block's position remains unchanged for a while, the planner will move the gripper to the other block and resume pushing until the task is finished.

**Planner in `Drawer-Opening`:** The planner selects a drawer randomly and tries to open it. If the chosen drawer fails to open after a while, the gripper will move to the other drawer and repeat the opening action.

## C.2 Network Architecture

The specific architecture used to learn representations is shown in Fig. 23.

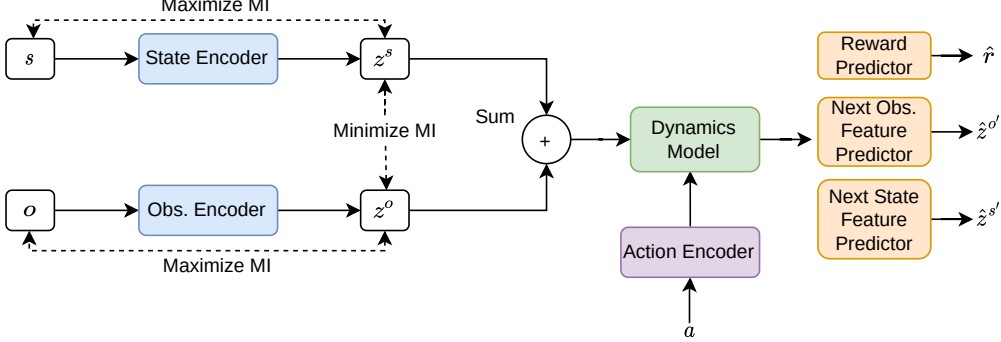

Figure 23: Architecture to learn representations in all domains.

Next, we describe the components for each domain from Fig. 24 to Fig. 27. To succinctly describe the network architecture, we use the following acronyms: **FC**($n$): a fully connected layer with $n$ outputs; **Conv**($f$, $s$): a convolutional layer with filter size $f \times f$ and stride $s$, **R** is the ReLU activation, and **MaxPool**($w$): a max pooling layer with window size $w$.

## C.3 Mutual Information Estimation

### C.3.1 Minimizing $I(z^s; z^o)$

From the upper bound equation Eq. (2), we minimize its variational estimate defined below:

$$\mathcal{L}_{\text{CLUB}} = \frac{1}{B^2} \sum_{i=1}^{B} \sum_{j=1}^{B} [\log q(z_i^o | z_i^s) - \log q(z_j^o | z_i^s)] \tag{7}$$

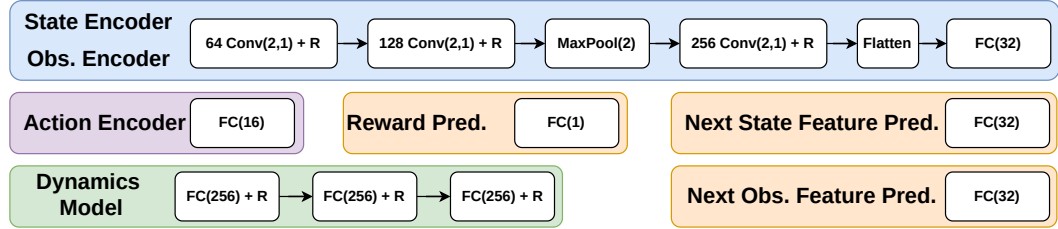

Figure 24: Network architecture in `Sphinx`.

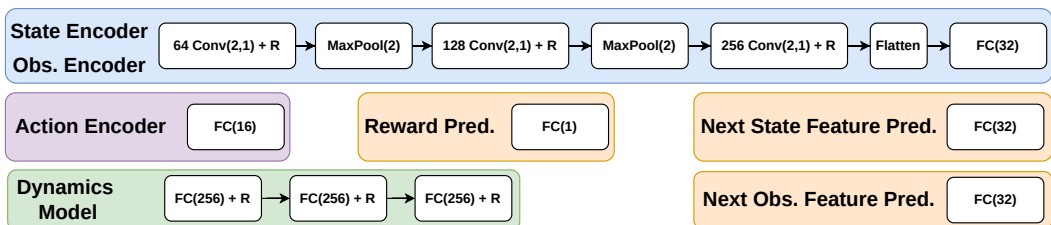

Figure 25: Network architecture in `CarFlag-2D`.

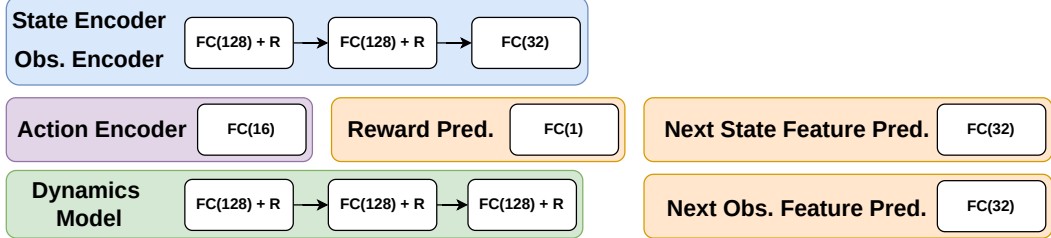

Figure 26: Network architecture in `Heaven-Hell`.

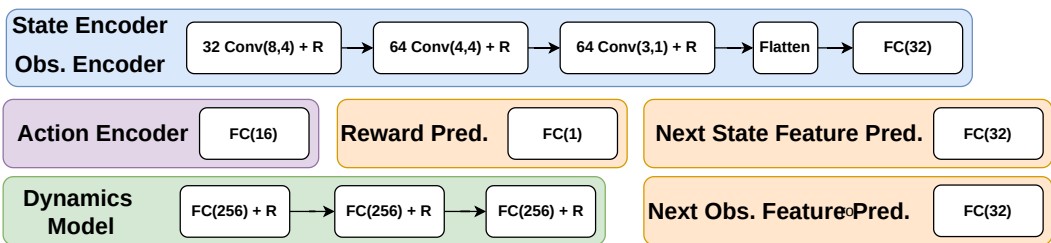

Figure 27: Network architecture in robot domains.

The variational distribution $q(z^o|z^s)$ is updated to minimize $\mathbb{D}_{\text{KL}}\left[q(z^o|z^s) \parallel p(z^o|z^s)\right]$. We assume $q(z^o|z^s)$ follows a Gaussian distribution and use the following network architectures:

**Mean network: FC(32) → R → FC(32)**

**Log variance network: FC(32) → R → FC(32) → Tanh**

We use the batch size $B = 500$ and use a learning rate of $0.001$ for all tasks, except `Heaven-Hell`, in which a learning rate of $0.0003$ is used. We update $q$ whenever we update $\phi(s)$ and $\psi(o)$.

### C.3.2 Maximizing $I(o; z^o)$ and $I(s; z^s)$

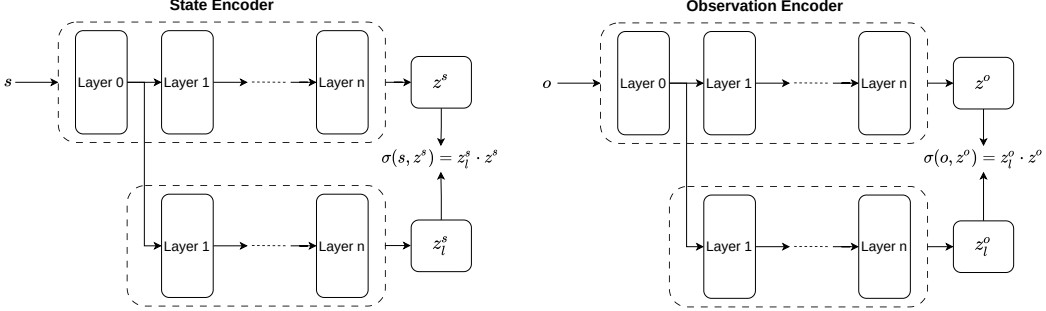

Figure 28: Architecture to calculate $\sigma(s, E(s))$ and $\sigma(o, E(o))$ using the dot product operation.

From the lower bound equation Eq. (3), we minimize the following loss:

$$\mathcal{L}_{\text{DIM}} = \frac{1}{B^2} \sum_{i=1}^{B} \sum_{j=1}^{B} [\text{sp}(-\sigma(x_i, E(x_i))) + \text{sp}(\sigma(x_j, E(x_i)))] \tag{8}$$

As shown in Fig. 28, the discriminator $\sigma$ uses the same architecture of the state encoder $\phi$ (when calculating the state feature $z_l^s$ of $s$) and the observation encoder $\psi$ (when calculating the observation feature $z_l^o$ of $o$). We dot product to compute $\sigma(s, E(s)) = z_l^s \cdot z^s$ and $\sigma(o, E(o)) = z_l^o \cdot z^o$.

### C.4 Hyper-parameters

We provide the hyper-parameters used for training representations in Table 3.

Table 3: Hyper-parameters used in learning representation. `HH`: `Heaven-Hell`, `S`: `Sphinx`, `CF`: `CarFlag-2D`, `BP`: `Block-Pulling`, `BPs`: `Block-Pushing`, and `DO`: `Drawer-Opening`.

| Domain | HH | CF | S | BP | BPs | DO |
|---|---|---|---|---|---|---|
| # of samples | 21785 | 45406 | 13682 | 1226 | 1240 | 1234 |
| # of episodes | 500 | 1000 | 500 | 80 | 80 | 80 |
| # of augmentations per sample | - | - | - | 4 | 12 | 6 |
| # of training epochs | 1000 | 1000 | 1000 | 1000 | 1000 | 1000 |
| Batch size $B$ | 500 | 500 | 500 | 500 | 500 | 500 |
| Learning rate | 0.003 | 0.001 | 0.001 | 0.001 | 0.001 | 0.001 |
| Reward loss coeff. $\lambda_r$ | 10.0 | 1.0 | 10.0 | 10.0 | 100.0 | 100.0 |
| State loss coeff. $\lambda_s$ | 1.0 | 1.0 | 0.5 | 0.1 | 1.0 | 1.0 |
| Observation loss coeff. $\lambda_o$ | 0.5 | 5.0 | 0.03 | 1.0 | 1.0 | 1.0 |
| $\downarrow I(z^s; z^o)$ loss coeff. $\lambda_{\text{CLUB}}$ | 1.0 | 10.0 | 0.3 | 10.0 | 0.001 | 1.0 |
| $\uparrow I(s; z^s)$ loss coeff. $\lambda_{\text{DIM}}$ | 0.0 | 0.0 | 0.0 | 0.1 | 0.01 | 0.001 |
| $\uparrow I(o; z^o)$ loss coeff. $\lambda_{\text{DIM}}$ | 1.0 | 1.0 | 0.5 | 1.0 | 1.0 | 1.0 |

# D   Additional Experiments

## D.1   Using $z^s \oplus z^o$ versus $z^s$ for Task Learning

Continuing the experiment from Section 5.2.1, we report the performance using $z^s$ and $z^s \oplus z^o$ for task learning in all domains.

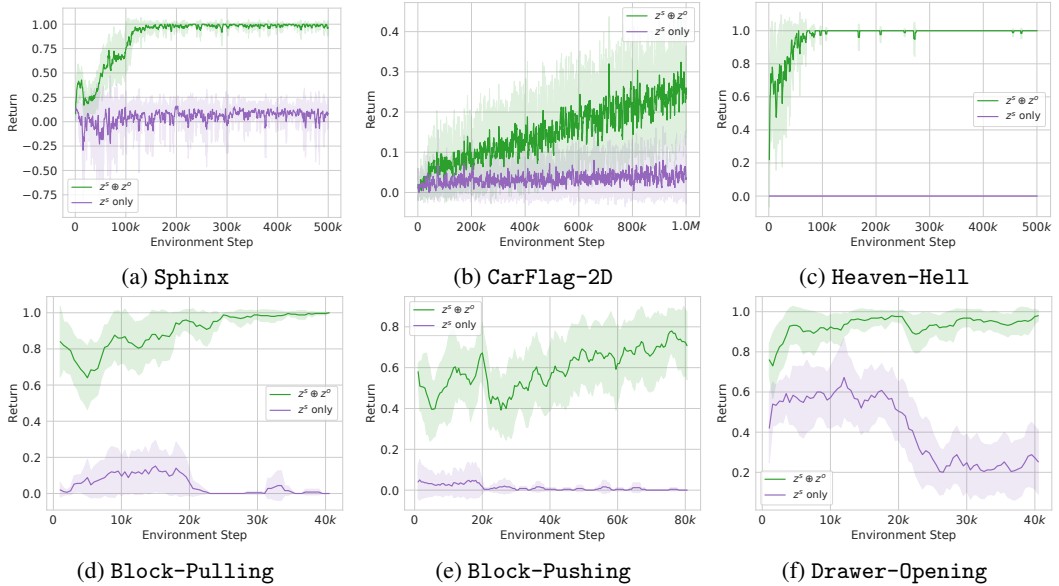

Figure 29: Task learning performance when using $z^s$ and $z^s \oplus z^o$ as the "state".

## D.2   Using Only Auxiliary Task/Intrinsic Rewards

Here, we show the learning performance when using intrinsic rewards and/or the auxiliary task.

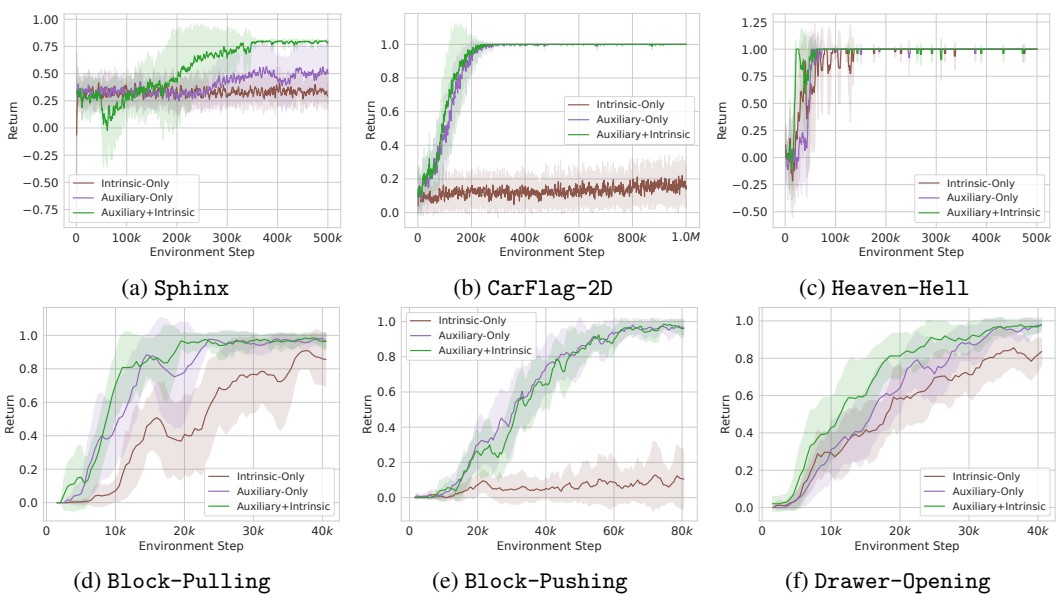

Figure 30: Comparing using intrinsic rewards or the auxiliary task versus using both.

## D.3 Using GRU v.s. GPT

Here, we report the performance in all domains when using a GRU versus GPT as the sequence model in our proposed agent.

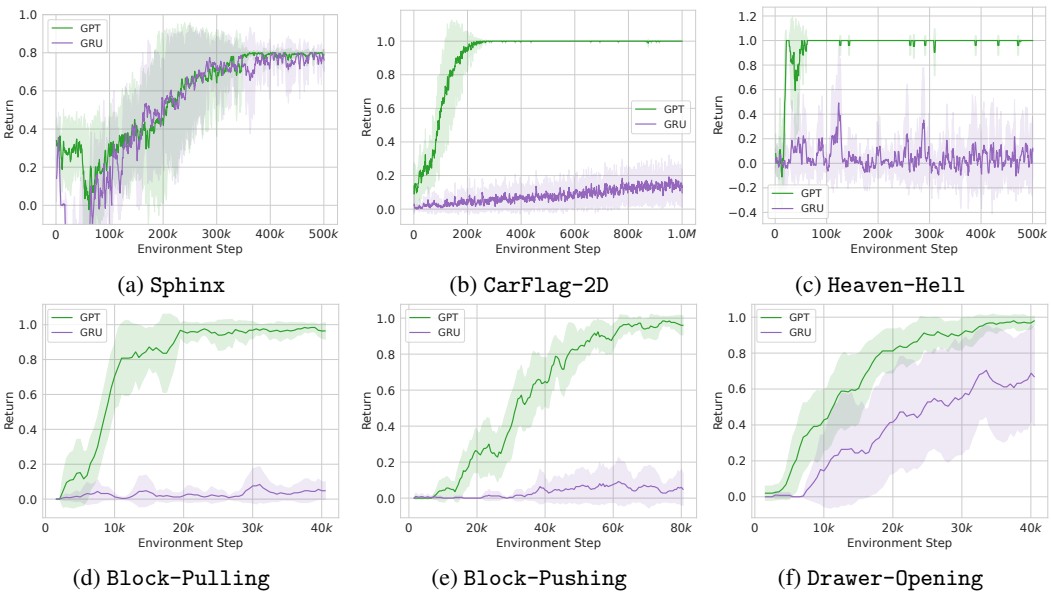

(a) Sphinx      (b) CarFlag-2D      (c) Heaven-Hell

(d) Block-Pulling      (e) Block-Pushing      (f) Drawer-Opening

Figure 31: Task learning performance when using a GRU v.s. GPT.

## D.4 Visualization of Intrinsic Rewards

We visualize the intrinsic rewards of trained agents in three grid-world domains in Fig. 32. The intrinsic rewards peak when the agents perform the information-gathering actions.

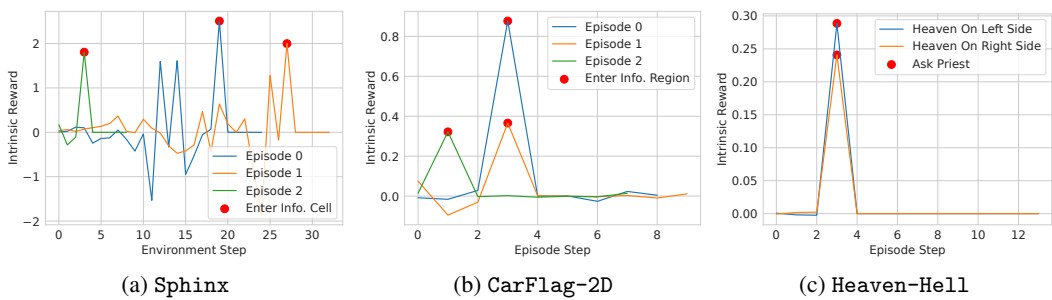

(a) Sphinx      (b) CarFlag-2D      (c) Heaven-Hell

Figure 32: Intrinsic rewards within an episode of trained agents in three grid-world domains. Red circles denote when the intrinsic rewards peak, e.g., when they perform informative actions.

## E Details of Hardware Experiments

### E.1 Obtaining Depth Images

We fuse the point clouds from two RealSense D455 cameras (Cam 1 and Cam 2) and one Azure Kinect camera (Cam 3) to create an integrated point cloud (see Fig. 33). We then orthographically project the point cloud at the gripper's position to create a depth image observation. Examples of observations in the three robot domains can be seen in Fig. 34.

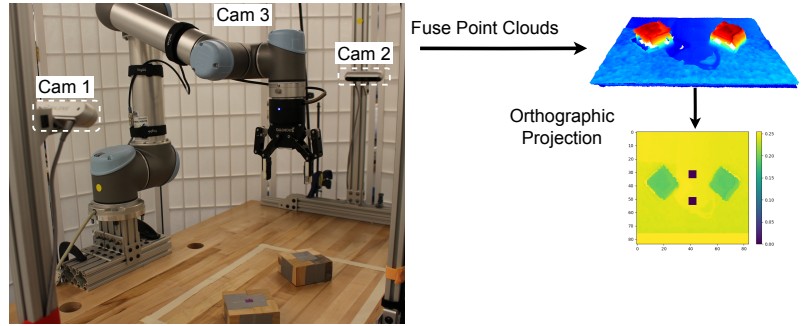

Figure 33: We fuse the point clouds from three cameras (to avoid occlusions) and performed an orthographic projection at the gripper's position to create a depth image observation.

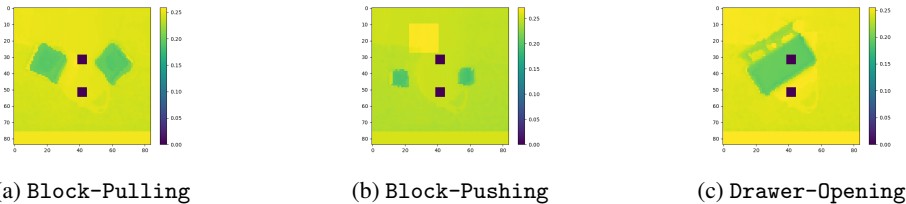

(a) `Block-Pulling`            (b) `Block-Pushing`            (c) `Drawer-Opening`

Figure 34: Examples of observations in real robot experiments.

## E.2 Added Perlin Noise for Better Sim-To-Real Transfers

Following [39], we found it useful for better sim-to-real transfers by adding the Perlin [46] noise to the depth images during training for more robust policies by being closer to real-world depth images. For all robot domains, we applied the noise with a magnitude of 7mm (see Fig. 35).

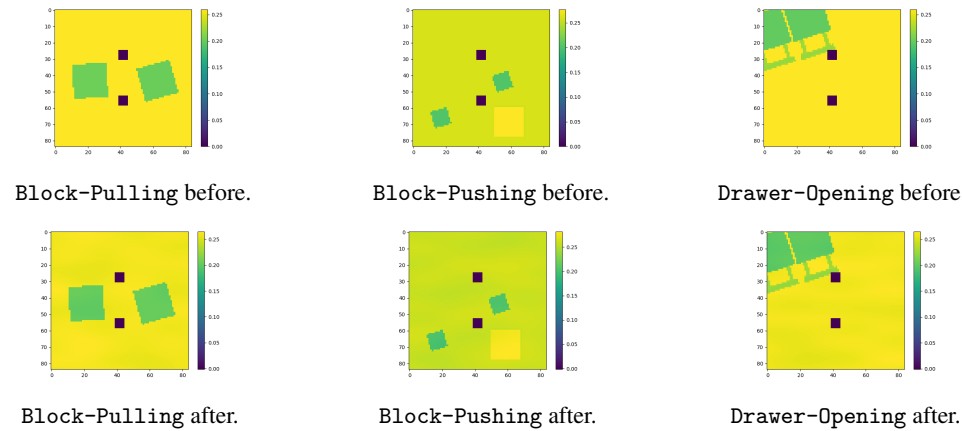

`Block-Pulling` before.          `Block-Pushing` before.          `Drawer-Opening` before.

`Block-Pulling` after.          `Block-Pushing` after.          `Drawer-Opening` after.

Figure 35: Depth images before and after adding Perlin [46] noise for better sim-to-real transfers.

## F  Details of SO(2) Rotational Data Augmentation

We perform SO(2) rotational augmentation by choosing a random angle and rotating the depth images around its center. We perform this augmentation in two cases:

**When learning the representations to utilize the data better.** For each transition $(s, o, a, r, s', o')$, we sample a random angle and rotate $s, o, s', o'$ at the same angle. Each transition has its own random angle, see Fig. 36 for examples.

**When performing task learning robot domains.** Given an episode, we first sample a random angle and apply the rotation with this angle for *every* $s, o, s', o'$ within the episode. Because we are trying to learn a history-based policy, this is to ensure the augmented history is valid (see Fig. 37).

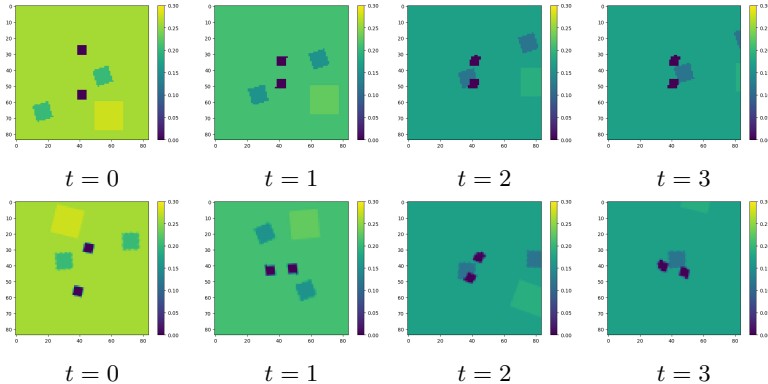

Figure 36: Examples of rotation data augmentation applied for transitions in an episode in `Block-Pushing` to augment the data for learning the representation: a *different* random rotation is applied independently for $s, o, s', o'$ in each timestep in an episode.

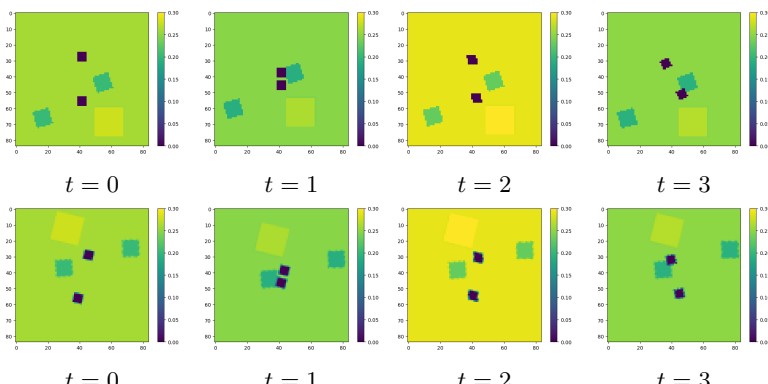

Figure 37: Examples of rotation data augmentation applied for an episode in `Block-Pushing`: the *same* random rotation is consistently applied to every tuple $s, o, s', o'$ within an episode.

