# OpenReview forum: "Leveraging Mutual Information for Asymmetric Learning under Partial Observability"
_robot-learning.org/CoRL/2024/Conference — CoRL 2024_

### Official Review · Reviewer_VPtp · 2024-07-18

**Originality:** 3
**Technical Quality:** 3
**Clarity Of Presentation:** 4
**Potential Impact:** 3
**Recommendation:** 3
**Confidence:** 3

**Review:**

Quality: The paper is overall of a high quality. The writing and figures are well done, and the theory (albeit limited) is correct and well explained.

Clarity: The paper is cleanly written with minimal jargon and no unnecessary formalism. The extensive appendix is useful to get information on the experimental setup which is slightly terse in the main paper.

Originality and Significance: The use of privileged information for policy learning is relatively well established in the literature. However, the paper makes a meaningful contribution by showing improvements across a wide range of baselines on some tasks. I think a full evaluation of the potential impact is difficult as the set of tasks is somewhat limited and the performance improvements are not fully clear

## Open issues

Overall, the idea seems to be well supported in the experiments. However, I believe that the ablations are incomplete and deserve more attention. The introduced method changes several details compared to the compared baselines and the source of the improvements are unclear.

The ablation clearly shows that a majority of the performance of the method seems to stem from the use of a GPT style transformer as opposed to the GRU cells, with the intrinsic reward also contributing some performance. However, i.e. the DTQN baseline is stated to use DQN while the proposed method uses DDQN. Since this is an orthogonal issue, I would ask the authors to rerun ablations with a more clear and equal footing. Furthermore, ZP-DRQN etc are claimed to require a GRU, but I am not convinced that this is true. While Ni et al state that transformers do not fulfill the recurrency condition, I am not convinced that the test environments require a full recurrent state encoder. The baseline comparison/ablation requires substantial care here, and the quoted paper talks about a more general set of problems (i.e. POMDPs with in principle unbounded history memory requirements) which the use of transformers in this method would be unable to solve.

I want to highlight that the authors were otherwise very careful with the baseline comparison, so this is a bit nipicky.

This problem also applies to the intrinsic reward. I would like to ask the authors to comment on which baseline methods could be augmented with the same reward. I think it is important to not label arbitrary combinations of orthogonal advancements as their own "algorithms" in reinforcement learning, but to embrace the "rainbow lesson" and evaluate heuristics and improvements as individual improvements which can be freely combined.

The robotics experiments are somewhat short, given the focus of the conference, I am unsure whether the method is best placed at CoRL. However, since the authors do include both some arguments for why the problem is relevant for robotics and some real world experiments, I am not weighing this concern strongly.

**Quality Of The Limitations Section:**

2

**Questions For Rebuttal:**

As a simple additional baseline, I would ask the authors to run their method _without_ a history (or only a very short history) to highlight how much benefit comes from being able to aggregate information across multiple timesteps and how much is simply due to the architectural improvements.

I would ask the authors to repeat the sim to real experiments with the strongest baselines as well to have a better basis for comparison.

I would like to see the issues discussed in the limitations section to be highlighted and discussed in more depth earlier in the paper, especially how it influences the choice of evaluation environment and whether some baselines might outperform the proposed method on suboptimal data.

**Robotics Focus:**

4

**Summary Of Paper:**

The paper proposes a novel method for learning from asymetric information in partially observable

**Summary Of Recommendation:**

The paper presents a convincing idea, with some minor issues and details in the experimental setup that seem easy to clarify in the rebuttal.

---

### Official Review · Reviewer_TSYf · 2024-07-20
**Review of Leveraging Mutual Information for Asymmetric Learning under Partial Observability**

**Originality:** 3
**Technical Quality:** 3
**Clarity Of Presentation:** 4
**Potential Impact:** 3
**Recommendation:** 3
**Confidence:** 4

**Review:**

Strengths:
- This paper introduces a novel exploration bonus that encourages information-seeking behaviors, by assuming access to ground-truth states during training.
- The method is demonstrated on a series of simulated tasks that require exactly such information-gathering behaviors, and it outperforms prior methods that do not have such exploration explicitly incorporated into them.
- Paper is written clearly.

Weaknesses:
- I don't think the paper is appropriately citing what is new in this paper and what has been introduced in prior work already. Specifically, the section on "Encouraging Information Seeking", which discusses the derivation of the reward bonus, is taken nearly directly from Liu et al [1]. I think it should be attributed accordingly. Additionally, I wanted to point out recent/concurrent work that's probably of interest to the authors. It introduces a similar exploration bonus for the same problem setting [2].
- Access to privileged state information even during training can be an overly strong and restrictive assumption.
- Due to this assumption, the method is probably most useful for scenarios where we can first train in simulation, which more readily offers such privileged information, and transfer to real.
- The current experimental domains require pretty simple information-gathering strategies, and could be casted as meta-RL problems. How would meta-RL methods like [1], [3], [4] compare here?

[1] Liu et al. Decoupling exploration and exploitation for meta-reinforcement learning without sacrifices. ICML 2021.

[2] Xie et al. Learning to Explore in POMDPs with Informational Rewards. ICML 2024.

[3] Zintgraf et al. Varibad: A very good method for bayes-adaptive deep RL via meta-learning. 2019.

[4] Kamienny et al. Learning adaptive exploration strategies in dynamic environments through informed policy regularization. 2020.

**Quality Of The Limitations Section:**

3

**Questions For Rebuttal:**

- How is the hyperparameter $\alpha_t$ chosen? Why is it a function of the episode time-step?
- What happens if the privileged state is incorrectly specified or is still a partial observation?
- How would meta-RL methods compare on these experimental domains?
- What are the main failure modes in the sim-to-real transfer experiment?

**Robotics Focus:**

2

**Summary Of Paper:**

This paper introduces an intrinsic reward bonus that encourages an RL agent to perform active information-gathering in partially-observable settings. The main idea is to assume access to the hidden states during training and introduce an exploration bonus to the policy for gathering information about these hidden states. The method is demonstrated on grid world problems and robotic manipulation tasks.

**Summary Of Recommendation:**

Overall, the paper presents a new method for learning exploration behaviors in partially-observable settings and demonstrates that it can indeed learn such behaviors in specific scenarios that call for them. However, some main flaws of the paper is that it doesn’t properly attribute certain derivations to prior work, and the experimental domains and exploration strategies required are quite simple.

---

### Official Review · Reviewer_p5c4 · 2024-07-24
**Review for Submission 256**

**Originality:** 3
**Technical Quality:** 3
**Clarity Of Presentation:** 2
**Potential Impact:** 3
**Recommendation:** 3
**Confidence:** 3

**Review:**

Strengths:
- The authors present a novel MI objective grounded by prior work.
- The authors show efficacy of the method across domains and show a substantial improvement over alternative approaches
- The authors show the transfer of their approach to real domains while training in sim.
- The authors perform a rigorous ablation study for the different design decisions considered

Weaknesses:
- Would be good to consider the sample complexity of an algorithm, given the first stage requiring a high quality dataset to train with as stated in the limitation section.
- The tasks considered seem slightly contrived and would be interesting to see how this approach works for more complicated domains
- The approach has many moving components which draws up concerns about scalability and efficacy when the data, architecture, and algorithm is scaled up with the objective functions designed.
- Can improve clarity of paper (e.g some work in figures 1,4) could be helpful for readers to better understand the approach.

**Quality Of The Limitations Section:**

3

**Questions For Rebuttal:**

- One limitation seems to be that the first stage needs high quality demonstations. It would be interesting to consider how an imitation learning baseline (e.g diffusion policy) does with these demos as a point of comparison. This could also be further fine-tuned with the reinforcement learning using some ablation over the intrinsic/extrinsic reward to compare with the representation learned in P1. To deal with the partial observability using some history (either observations/actions) may be needed.
- The discussion about the two branch architecture and Figure 1 are slightly unclear to understand how the concatenation is done and how it encourages the separation of $z^s$ and $z^0$. I understand that the objectives (e.g minimizing MI) can lead to this property but it isn't clear to me why the architecture choice can lead to this. Could you please clarify this?

**Robotics Focus:**

4

**Summary Of Paper:**

Work provides a MI-based Objective to encourage exploration in partially observable situations.

**Summary Of Recommendation:**

Provide a novel approach that outperforms baselines presented. Some issues with clarity and potential lack of scalabilty.

---

### Author Rebuttal · Authors · 2024-08-07

We thank all reviewers for constructive reviews to improve our paper. Besides addressing all reviews below, we also attach a zip file, which will include the revised paper and pdf responses to each reviewer (to be updated).

The current zip file includes:
- Response_Reviewer_p5c4.pdf
- Response_Reviewer_VPtp.pdf
- Reponse_Reviewer_TSYf.pdf

---

### Decision · Program_Chairs · 2024-09-04

**Decision:**

Accept

**Comment:**

Pre-rebuttal:
This paper received 3 fairly confident reviews. The paper tackles learning under PO; they propose using state-observation and state-history mutual information; they use privileged training mechanisms and perform an ablation study for the different design decisions considered.

To address in rebuttal:
- Discuss sample complexity
- Address the nature of the chosen tasks (they seem contrived)
- Address limitations of imitation learning approach
- Improve on clarity (see reviewer comments)
- Better situate the work (eg, within Meta learning)
- Discuss the robustness of the current sim-to-real transfer strategy

Post-rebuttal:
The authors have addressed the issues pointed out by reviewers, and provided an improved manuscript; in particular, they have added experiments that further clarify the contributions of the method, juxtapose it more clearly with state of the art baselines, as well as the sim-to-real transfer robustness.